# Knowledge, attitude and practice toward childhood immunization among mothers in Lebanon

Sara Saleh[1], Pia Chedid [2]*

1 Department of Nursing, Faculty of Health Sciences, Beirut Arab University, Beirut, Lebanon,
2 Department of Medical Laboratory Technology, Faculty of Health Sciences, Beirut Arab University, Beirut, Lebanon

* p.chedid@bau.edu.lb

## Abstract

### Background

Barriers to childhood immunization in Lebanon are multifaceted. While Lebanon has historically maintained high immunization coverage, recent socio-economic challenges and recurrent political crisis have strained the healthcare system, leading to decreased vaccination rates. The knowledge, attitudes, and practices (KAP) of mothers play a key role in determining whether their children receive recommended immunizations. The aim of this study was to assess the KAP of Lebanese mothers regarding childhood immunization and to identify potential barriers to vaccination in children.

### Methods

A cross-sectional study was conducted among Lebanese mothers with children aged between zero days to 5 years old. A stratified random sampling method was used and a structured questionnaire was administered to collect data on KAP regarding childhood immunization. Descriptive statistics were performed. Chi-square test and linear regression were used to assess the association between KAP and sociodemographic factors with significance set at $p < 0.05$.

### Results

Most mothers demonstrated good knowledge (86.3%) and positive attitudes (94.8%) toward vaccination. However, only 43.3% exhibited good vaccination practices. Significant disparities were observed in rural regions such as Beqaa, where mothers had the lowest education levels, the highest rates of poor knowledge (11.9%), and the lowest vaccination practice scores (32.7%). Employment was significantly associated with higher knowledge scores (p = 0.002, 95%

**Data availability statement:** All relevant data are within the paper and its Supporting information files.

**Funding:** The author(s) received no specific funding for this work.

**Competing interests:** The authors have declared that no competing interests exist.

CI: -1.42 to -0.33), while household income was significantly associated with attitudes (p = 0.009, 95% CI: -0.96 to -0.13) and vaccination practices (p = 0.002, 95% CI: -0.48 to -0.11). Vaccine unavailability and cost were frequently cited as barriers, particularly in rural regions.

## Conclusions

Discrepancy between knowledge and attitude v/s practice suggests that a number of barriers prevent mothers from translating their good knowledge and positive attitude into action. Significant regional disparities exist, with rural areas like Beqaa facing the greatest challenges. Strengthening vaccine accessibility, implementing targeted educational interventions, and increasing governmental support are critical to improving immunization coverage nationwide.

---

## Introduction

The Under Five Mortality Rate (U5MR) refers to the probability that an infant will die before the age of five. It is expressed as the number of deaths of children under five years-old per 1,000 live births. According to the World Health Organization (WHO), the leading causes of death in children under the age of five are pneumonia, diarrhea and malaria, as well as complications of prematurity and birth asphyxia/trauma [1].

The U5MR was found to be on average 64% lower in a country whose entire population has received a combined diphtheria, tetanus, and pertussis vaccine (DPT3) than in a country with no DPT3 coverage [2]. The concept of vaccination is not limited to one person; it affects entire communities [3]. Vaccinated children can protect themselves and others by preventing transmission of vaccine-preventable diseases (VPDs) [3]. Between 2010 and 2018, 23 million deaths were averted with measles vaccine alone [4]. Maternal education significantly influences child health outcomes, as educated mothers are more likely to understand the importance of vaccinations, thereby improving immunization coverage and reducing U5MR [5,6]. In Lebanon, the U5MR in 2021 was estimated at 8 deaths per 1,000 live births. This compares well with the U5MR in the Arab world with 34 deaths per 1,000 live births and the U5MR in low- and middle-income countries (LMIC) with 42 deaths per 1000 live births [1].

Successful immunization programs depend on adequate coverage to maintain herd immunity [7]. However, several socioeconomic factors may affect immunization coverage. For example, low socioeconomic status leads to opposite practical circumstances, such as the lack of transportation, which helps prevent completion of vaccination. The acceptance of a program also depends on the parents' knowledge and attitude towards vaccination [8]. In addition, the fear of adverse effects negatively affects vaccination coverage [9]. For example, in response to a hypothetical link to autism, measles, mumps and rubella (MMR), it was shown that vaccination coverage in parts of Scotland had fallen dramatically [10]. In Lebanon, an outbreak of measles

in 2013 and a strong increase in cases of mumps in 2015 are reminders of the vulnerability of the current Lebanese vaccination coverage system [11].

Beirut and Mount Lebanon are characterized by high population densities and urbanization, serving as the economic and cultural hubs of the country. These regions benefit from advanced infrastructure and higher socioeconomic levels, which attract internal migration from less developed areas [12]. In contrast, the Beqaa and South Lebanon are predominantly rural, with lower population densities and economies that heavily rely on agriculture. The Beqaa region is particularly noted for its fertile lands, while South Lebanon is recognized for its olive and citrus farming [13]. In rural areas, such as the Beqaa region, educational opportunities for women are often more limited compared to urban centers. Factors such as poverty, cultural norms, and lack of access to quality educational institutions contribute to lower educational attainment among women in these regions [14]. The educational level of parents and the socio-economic background of the family play a crucial role in shaping the decision-making process regarding immunization and, as a result, impact the rate at which immunizations are received [15]. Regarding the specific role of mothers in assuring children receive vaccination, many studies conclude that mothers with a higher level of education tend to vaccinate more their children [16–18]. Mothers' self-efficacy and confidence in making vaccination decisions have also been shown as critical predictors of adherence to vaccination schedules [19]. In fact, when mothers are well-informed and supportive of vaccination, they are more likely to ensure their children receive timely immunizations, thereby reducing the risk of vaccine-preventable diseases (VPDs) [20]. In Lebanon, very few studies have assessed the barriers to toward childhood immunization, especially in the current context of the socioeconomic crisis the country is facing [21]. Notably, a study published in 2020 revealed that effective parent-physician communication was associated with better vaccination practices, while higher education and trust in healthcare professionals positively impacting attitudes and adherence to immunization schedules [22]. More recently, a study on human papillomavirus (HPV) vaccination in Lebanon found that only 12.8% of children were vaccinated, with willingness strongly influenced by parental awareness, prior HPV vaccination, and insurance coverage, while cost and misinformation remained major obstacles [23].

In Lebanon, the compounded effects of the economic collapse, the COVID-19 pandemic, and the Beirut port explosion have led to a deterioration in health-seeking behaviors, with families prioritizing urgent care over preventive services, thereby increasing the risk of communicable diseases among children [11,24,25]. Reports indicate that a large portion of the Lebanese population has been facing budget constraints related to health expenses, which has resulted in self-medication and decreased access to essential medications for chronic conditions, further exacerbating health risks for vulnerable populations, including children [26].

Additionally, Lebanon's routine immunization calendar has undergone several revisions to address the public health needs of its population, particularly in light of the ongoing challenges posed by the Syrian refugee crisis and the COVID-19 pandemic. In Lebanon, the current immunization schedule includes a series of vaccines administered at specific ages, with notable vaccines such as the measles, mumps, and rubella (MMR) vaccine being administered in two doses and the Rota vaccine introduced in 2022 [27,28].

Despite these structured timelines, Lebanon faces significant barriers that hinder the effective implementation of its vaccination programs. One of the primary barriers to immunization in Lebanon is the socio-economic context, particularly the rising poverty levels exacerbated by the influx of refugees since 2011 [25]. This demographic shift has resulted in many children missing routine vaccinations, leading to concerns about herd immunity and the resurgence of VPDs [16,29]. Maternal KAP play a crucial role in childhood immunization uptake, yet misinformation, cultural beliefs, and access barriers can hinder vaccine acceptance. Studies on immunization KAP in Lebanon have primarily focused on specific regions, particularly urban areas such as Beirut and Mount Lebanon. Our study aims to address potential regional disparities and to inform targeted interventions to improve children immunization coverage. Using a cross-sectional design and a standardized questionnaire, KAP of Lebanese mothers residing in Beqaa, South, Mount Lebanon and Beirut regions was determined.

## Materials and methods

### Study design

A descriptive quantitative cross-sectional study was conducted during the period of July 2023 to September 2023. A total of 212 questionnaires, available in both Arabic and English, were distributed in various Lebanese districts, including Beqaa, Beirut, South, and Mount Lebanon.

### Ethical approval

Informed consent was obtained from each participant. The Institutional Review Board (IRB) of Beirut Arab University (BAU) approved the study protocol under code 2023-H-0152-HS-M-0542.

### Participant enrollment

The sample size was determined using Cochran's formula for cross-sectional studies:

$$Z^2 p \left(1 - p\right) / e^2 = 196$$

Where Z is the Z-score for a 95% confidence level (1.96), p is the estimated proportion of mothers with good knowledge of childhood immunization (assumed at 50% to maximize sample size), e is the margin of error, set at 7% (0.07).

To ensure representative geographic distribution, a stratified random sampling approach was used. Lebanon is divided into multiple administrative regions. For this study, mothers were recruited from Beirut, Mount Lebanon, Beqaa, and South Lebanon to reflect urban-rural disparities. The distribution was as follows: Mount Lebanon (n = 45), Beqaa (n = 101), Beirut (n = 47), and South Lebanon (n = 19). Participants in each region were randomly selected from healthcare centers, public health clinics, and community outreach programs to ensure unbiased selection.

### Data collection method

Lebanese mothers with children aged between zero days to 5 years old were invited to participate in the study. Trained personnel conducted in-person interviews. Inclusion criteria required participants to be Lebanese mothers with at least one child aged between zero to five years old. Participants needed to have the ability to comprehend and respond to the survey questions in either Arabic or English. Exclusion criteria encompassed mothers that were not Lebanese or not living in Lebanon, who did not have children within the specified age group, those who were unwilling to participate, or lacked the capability to understand or communicate in Arabic or English.

### Questionnaire descriptive

The questionnaire utilized in this study was available in both Arabic and English languages. It was developed by drawing upon items from a previous study by Almutairi et al., specifically focusing on the assessment of Mothers' KAP regarding childhood vaccination during the first five years of life [18]. The full questionnaire is available as supporting information (S1 Fig).

### Structure of the questionnaire

The section Sociodemographic Status of Lebanese Mothers (7 items) encompassed questions regarding the sociodemographic characteristics of Lebanese mothers, including age, educational level, job status, monthly income, number of children, age of children, and smoking habits. In this section, "literate" was defined as the ability to read and write simple sentences in any language without assistance. Participants categorized as "literate" had no formal education but were able to comprehend written material and complete the questionnaire independently. This category was distinct from "uneducated," which referred to individuals who had never attended school and reported being unable to read or write.

The section Knowledge Assessment (12 items) is the second part of the questionnaire aimed to assess the knowledge of Lebanese mothers regarding vaccination. Additionally, it explored the sources of their knowledge and included questions related to healthy lifestyle and its maintenance.

The section Attitude Assessment (9 items) delved into the attitudes of mothers towards vaccination and healthy lifestyle practices. It covered topics such as confidence in vaccination, opinions on natural immunity, distinctions between dispensaries and private clinics, and breastfeeding.

The section Practice Assessment (13 items) evaluated the vaccination practices of mothers. It encompassed questions related to whether mothers were up to date with vaccinations, instances of vaccine refusal or missed vaccinations, experiences of vaccine-related side effects, the source of vaccination fees, and whether the cost and unavailability of vaccines posed obstacles to vaccination.

### Scoring and categorization

A 3-point Likert scale was employed in the questionnaire, offering response options including "agree" "uncertain" and "disagree" each associated with scores ranging from 1 to 3 for each question. In the practice section, response options included "yes" and "no". Total KAP scores were calculated by adding scores of all answers from each section. To categorize responses into poor, moderate, and good, the visual binning tool in SPSS was used. The cut-off points were determined based on the percentile distribution of scores.

### Statistical analysis

The statistical analysis for this study was conducted using SPSS version 24 for Windows software. Descriptive statistics summarized categorical variables as frequencies and percentages. Chi-square test was used to assess associations between categorical variables, while linear regression analysis examined the relationship between sociodemographic factors and KAP scores. Results were reported as standardized beta coefficients with 95% confidence intervals, and a p-value < 0.05 was considered statistically significant.

### Results

Out of 212 participants, most mothers (34%) were aged 31 to 35 years old, followed by mothers aged 25 to 31 years (25%) and mothers over 35 years old (24.5%) (Table 1). Concerning mother's education, job and income, variations between regions were statistically significant (Table 1). Results showed that 50% of participants held a bachelor's degree or higher, with the highest percentage of educated mothers found in Mount Lebanon region (80%) (p = 0.001). On the other hand, mothers from the Beqaa region had the lowest percentage of bachelor's degree or higher (27.7%) (p = 0.001) (Table 1). Concerning mothers' occupation, most mothers were either employed (49.5%) or housewives (47.2%) (Table 1). Regarding monthly income, nearly half of the mothers (46.2%) indicated that their income was not enough, with this issue being most prominent in Mount Lebanon (60%) and Beqaa (55.5%) (p = 0.001) (Table 1).

Regarding family size, most mothers had 2–3 children (48.6%), with a substantial number having only one child (35.8%). Families with more than 3 children were less common in the general population (15.6%) but were the most represented in Beqaa region (21.8%) (Table 1). Concerning the age of children, most children were aged between 2–5 years (56.1%). Infants aged 0–6 months represented the smallest age group (13.2%). However, variation of family size and age of children according to region were not statistically significant (Table 1).

Mothers across all regions exhibited good knowledge and attitude scores, with respectively 86.3% and 94.8% of the total sample falling into this category (Table 2). A particularly high percentage of good knowledge score was reported in Mount Lebanon (95.6%) (Table 2). A small number of mothers had poor knowledge scores, making up 8.5% of the sample. This issue was most prominent in Beqaa and South regions, with 11.9% and 10.5% of mothers respectively having poor knowledge scores (Table 2). Poor attitude scores were also the highest in the Beqaa region (Table 2).

**Table 1. Socio-demographic characteristics of the population.**

| | General population (n=212) | Population per region | | | | Pearson Chi-square | p= |
|---|---|---|---|---|---|---|---|
| | n (%) | Mount Lebanon (n=45) | Beqaa (n=101) | Beirut (n=47) | South (n=19) | | |
| **Mother's age** | | | | | | | |
| 18–25 | 35 (16.5) | 8 (17.8) | 23 (22.8) | 3 (6.4) | 1 (5.3) | | |
| 25–31 | 53 (25) | 13 (28.9) | 22 (21.7) | 11 (23.4) | 7 (36.8) | | 0.306 |
| 31–35 | 72 (34) | 15 (33.3) | 31 (30.7) | 19 (40.4) | 7 (36.8) | 10.57 | |
| >35 | 52 (24.5) | 9 (20.0) | 25 (24.8) | 14 (29.8) | 4 (21.1) | | |
| **Mother's education** | | | | | | | |
| Uneducated | 4 (1.88) | 0 | 2 (2) | 2 (4.3) | 0 | | |
| Literate | 23 (10.84) | 3 (6.7) | 14 (13.9) | 4 (8.5) | 2 (10.5) | | |
| Primary School | 23 (10.84) | 2 (4.4) | 16 (15.8) | 5 (10.6) | 0 | 45.58 | 0.001* |
| Secondary School | 14 (6.6) | 0 | 12 (11.9) | 2 (4.3) | 0 | | |
| High School | 42 (19.81) | 4 (8.9) | 29 (28.7) | 6 (12.7) | 3 (15.8) | | |
| Bachelor or Higher | 106 (50) | 36 (80) | 28 (27.7) | 28 (59.6) | 14 (73.7) | | |
| **Mother's job** | | | | | | | |
| Student | 7 (3.3) | 1 (2.2) | 4 (4) | 2 (4.3) | 0 | | |
| Employee | 105 (49.5) | 39 (86.7) | 25 (24.7) | 27 (57.4) | 14 (73.7) | 56.32 | 0.001* |
| Housewife | 100 (47.2) | 5 (11.1) | 72 (71.3) | 18 (38.3) | 5 (26.3) | | |
| **Monthly income** | | | | | | | |
| Enough | 40 (18.9) | 0 | 17 (16.8) | 19 (40.4) | 4 (21.1) | | |
| Partly Enough | 74 (34.9) | 18 (40) | 28 (27.7) | 18 (38.3) | 10 (52.6) | 35.69 | 0.001* |
| Not Enough | 98 (46.2) | 27 (60) | 56 (55.5) | 10 (21.3) | 5 (26.3) | | |
| **Number of children** | | | | | | | |
| 1 | 76 (35.8) | 20 (44.4) | 33 (32.7) | 16 (34) | 7 (36.8) | | |
| 2–3 | 103 (48.6) | 23 (51.2) | 46 (45.5) | 23 (49) | 11 (57.9) | 9.29 | 0.158 |
| >3 | 33 (15.6) | 2 (4.4) | 22 (21.8) | 8 (17) | 1 (5.3) | | |
| **Children age** | | | | | | | |
| 0 to 6 months | 28 (13.2) | 3 (6.7) | 14 (14) | 8 (17) | 3 (15.8) | | |
| 6 to 12 months | 31 (14.6) | 9 (20.0) | 15 (14.9) | 5 (10.6) | 2 (10.5) | 4.38 | 0.884 |
| 12 to 24 months | 34 (16) | 9 (20.0) | 15 (14.9) | 7 (15) | 3 (15.8) | | |
| 2 to 5 years | 119 (56.1) | 24 (53.3) | 57 (56.4) | 27 (57.4) | 11 (57.9) | | |
| **Smoking status** | | | | | | | |
| Yes | 98 (46.2) | 22 (48.5) | 56 (55.4) | 29 (61.7) | 7 (36.8) | | |
| No | 114 (53.8) | 23 (51.5) | 45 (44.6) | 18 (38.3) | 12 (63.2) | 3.92 | 0.270 |

This table uses descriptive statistics: frequency (n) and percentage (%). Numbers with an asterisk correspond to significant associations.

The results of practice scores were more nuanced. Only 43.3% of mothers had a good practice score (Table 2). The percentage of mothers with poor practice score was relatively high in the general population (23.5%), with the highest representation in Beqaa (26.7%) and Beirut (25.5%) (Table 2). Only variations of practice scores according to regions were statistically significant (p=0.021) (Table 2).

In the knowledge section of the questionnaire, answers to two open-ended questions were analyzed (Figs 1 and 2). First, concerning the most common side effects of vaccination, the response rate was 98.6% (n=209). The most frequently reported side effect was "fever" mentioned by 40.7% (n=85), while 26.8% (n=56) of mothers reported no side

**Table 2. Knowledge, attitude and practice scores in the general population and according to regions.**

| | General population (n=212) | Population per region | | | |
|---|---|---|---|---|---|
| | n (%) | Mount Lebanon (n=45) | Beqaa (n=101) | Beirut (n=47) | South (n=19) |
| **Knowledge** | | | | | |
| Good | 183 (86.3) | 43 (95.6) | 84 (83.3) | 40 (85.1) | 16 (84.2) |
| Average | 11 (5.2) | 2 (4.4) | 5 (5) | 3 (6.4) | 1 (5.3) |
| Poor | 18 (8.5) | 0 | 12 (11.9) | 4 (8.5) | 2 (10.5) |
| **Attitude** | | | | | |
| Good | 201 (94.8) | 44 (97.8) | 94 (93) | 45 (95.7) | 18 (94.7) |
| Average | 6 (2.8) | 0 | 4 (4) | 1 (2.1) | 1 (5.3) |
| Poor | 5 (2.4) | 1 (2.2) | 3 (3) | 1 (2.1) | 0 |
| **Practice** | | | | | |
| Good | 92 (43.3) | 27 (60) | 33 (32.7) | 24 (51.1) | 8 (42.1) |
| Average | 70 (33.2) | 9 (20) | 41 (40.6) | 11 (23.4) | 9 (47.4) |
| Poor | 50 (23.5) | 9 (20) | 27 (26.7) | 12 (25.5) | 2 (10.5) |

This table uses descriptive statistics: frequency (n) and percentage (%).

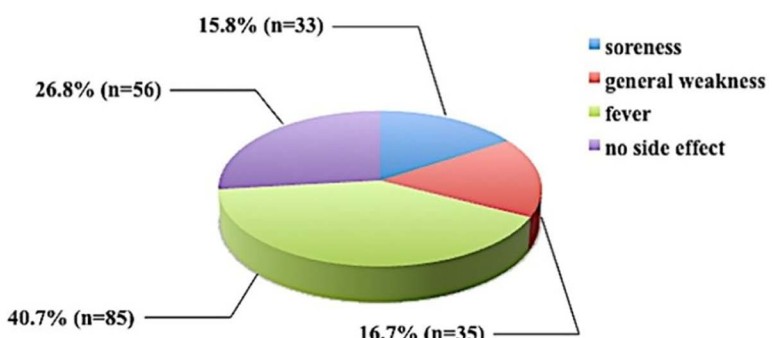

**Fig 1. Side effects of vaccination that affected participants' children.**

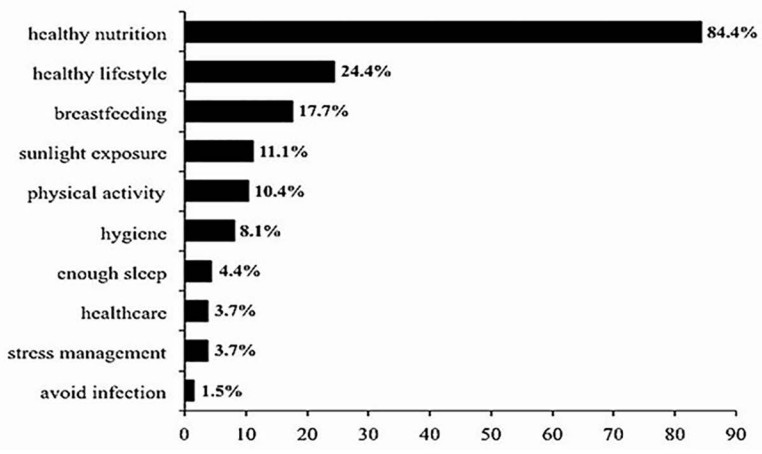

**Fig 2. Participants' perceived ways to boost children's immune system other than vaccination.**

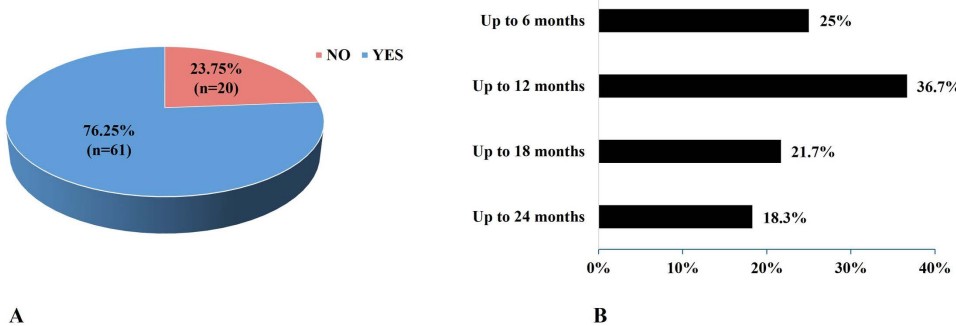

**Fig 3. Status (A) and duration (B) of breast feeding among participants.**

**Table 3. Bivariate analysis of sociodemographic and other factors associated with the knowledge scores.**

| n=212 | Knowledge | | | Pearson Chi-square | p= |
|---|---|---|---|---|---|
| | Good n (%) | Average n (%) | Poor n (%) | | |
| **Mother's age** | | | | | |
| 18 to 25 (n=35) | 33 (94.3) | 1 (2.9) | 1 (2.9) | 10.27 | 0.114 |
| 25 to 31 (n=53) | 48 (90.6) | 1 (1.9) | 4 (7.5) | | |
| 31 to 35 (n=72) | 57 (79.2) | 8 (11.1) | 7 (9.7) | | |
| 35 or older (n=52) | 45 (86.5) | 1 (1.9) | 6 (11.5) | | |
| **Number of children** | | | | | |
| One | 70 (92.1) | 1 (1.3) | 5 (6.6) | 14.63 | 0.006* |
| Two to three | 91 (88.3) | 6 (5.8) | 6 (5.8) | | |
| More than three | 22 (66.7) | 4 (12.1) | 7 (21.2) | | |
| **Mother's education** | | | | | |
| Uneducated | 2 (50) | 0 | 2 (50) | 28.11 | 0.002* |
| Literate | 20 (87) | 1 (4.3) | 2 (8.7) | | |
| Primary school | 14 (60.9) | 3 (1.3) | 6 (26.1) | | |
| Secondary school | 11 (78.6) | 1 (7.1) | 2 (14.3) | | |
| High school | 37 (88.1) | 3 (7.1) | 2 (4.8) | | |
| Bachelor or higher | 99 (93.4) | 3 (2.8) | 4 (3.8) | | |
| **Mother's job** | | | | | |
| Student | 5 (71.4) | 2 (28.6) | 0 (0) | 16.14 | 0.003* |
| Employee | 98 (93.3) | 2 (1.9) | 5 (4.8) | | |
| Housewife | 80 (80) | 7 (7) | 13 (13) | | |
| **Monthly income** | | | | | |
| Enough | 35 (87.5) | 1 (2.5) | 4 (10) | 3.75 | 0,440 |
| Partly enough | 66 (89.2) | 5 (6.7) | 3 (4.1) | | |
| Not enough | 82 (83.7) | 5 (5.1) | 11 (11.2) | | |
| **Smoker** | | | | | |
| Yes | 88 (89.8) | 1 (1) | 9 (9.2) | 6.46 | 0.040* |
| No | 95 (83.3) | 10 (8.8) | 9 (7.9) | | |
| **Region** | | | | | |
| Mount Lebanon | 43 (95.6) | 2 (4.4) | 0 | 6.05 | 0.418 |
| Beqaa | 84 (83.2) | 5 (4.9) | 12 (11.9) | | |
| Beirut | 40 (85.1) | 3 (6.4) | 4 (8.5) | | |
| South | 16 (84.2) | 1 (5.3) | 2 (10.5) | | |

This table uses descriptive statistics: frequency (n) and percentage (%). Numbers with an asterisk correspond to significant associations.

**Table 4. Bivariate analysis of sociodemographic and other factors associated with the attitude scores.**

| n=212 | Attitude | | | Pearson Chi-square | p= |
|---|---|---|---|---|---|
| | Good n (%) | Average n (%) | Poor n (%) | | |
| **Mother's age** | | | | | |
| 18 to 25 (n=35) | 34 (97.1) | 0 (0) | 1 (2.9) | 12.65 | 0.049* |
| 25 to 31 (n=53) | 52 (98.1) | 1 (1.9) | 0 (0) | | |
| 31 to 35 (n=72) | 68 (94.4) | 4 (5.6) | 0 (0) | | |
| 35 or older (n=52) | 47 (90.4) | 1 (1.9) | 4 (7.7) | | |
| **Number of children** | | | | | |
| One | 74 (97.4) | 1 (1.3) | 1 (1.3) | 3.47 | 0.480 |
| Two to three | 97 (94.2) | 4 (3.9) | 2 (1.9) | | |
| More than three | 30 (90.9) | 1 (3) | 2 (6.1) | | |
| **Mother's education** | | | | | |
| Uneducated | 3 (75) | 0 | 1 (25) | 15.69 | 0.100 |
| Literate | 23 (100) | 0 | 0 | | |
| Primary school | 21 (91.3) | 1 (4.3) | 1 (4.3) | | |
| Secondary school | 12 (85.7) | 1 (7.1) | 1 (7.1) | | |
| High school | 41 (97.6) | 0 | 1 (2.4) | | |
| Bachelor or higher | 101 (95.3) | 4 (3.8) | 1 (0.9) | | |
| **Mother's job** | | | | | |
| Student | 6 (85.7) | 0 | 1 (14.3) | 5.24 | 0.260 |
| Employee | 99 (94.3) | 4 (3.8) | 2 (1.9) | | |
| Housewife | 96 (47.8) | 2 (2) | 2 (2) | | |
| **Monthly income** | | | | | |
| Enough | 39 (97.5) | 0 | 1 (2.5) | 1.67 | 0.790 |
| Partly enough | 69 (93.2) | 3 (4.1) | 2 (2.7) | | |
| Not enough | 93 (94.9) | 3 (3.1) | 2 (2) | | |
| **Smoker** | | | | | |
| Yes | 92 (93.9) | 2 (2) | 4 (4.1) | 2.71 | 0.250 |
| No | 109 (95.6) | 4 (3.5) | 1 (0.9) | | |
| **Region** | | | | | |
| Mount Lebanon | 44 (97.8) | 0 | 1 (2.2) | 2.91 | 0.820 |
| Beqaa | 94 (93.1) | 4 (4) | 3 (3) | | |
| Beirut | 45 (95.7) | 1 (2.1) | 1 (2.1) | | |
| South | 18 (94.7) | 1 (5.3) | 0 | | |

This table uses descriptive statistics: frequency (n) and percentage (%). Numbers with an asterisk correspond to significant associations.

effect (Fig 1). The next question determined the mothers' knowledge about how to boost a child's immune system other than vaccination. The response rate to this question was lower: 63.6% (n=135). The most common response was "healthy nutrition" (84.4%, n=114), followed by "healthy lifestyle" (24.4%, n=33) and "breastfeeding" (17.7%, n=29) (Fig 2). To understand if these beliefs were translated into practice, we analyzed the response of 81 mothers regarding breast-feeding practice and duration of breastfeeding. Results showed that 76.25% (n=61) of mothers did follow breastfeeding (Fig 3A). The most common breastfeeding duration was 6–12 months (36.7%, n=22) (Fig 3B).

Using bivariate analyses, we determined the variation of knowledge (Table 3), attitude (Table 4) and practice (Table 5), according to other variables pertaining to the participants. We found a significantly higher knowledge score in mothers

**Table 5. Bivariate analysis of sociodemographic and other factors associated with the practice scores.**

| n=212 | Practice | | | Pearson Chi-square | p= |
|---|---|---|---|---|---|
| | Good n (%) | Average n (%) | Poor n (%) | | |
| **Mother's age** | | | | | |
| 18 to 25 (n=35) | 19 (54.3) | 11 (31.4) | 5 (14.3) | 5.84 | 0.441 |
| 25 to 31 (n=53) | 21 (39.6) | 19 (35.8) | 13 (24.5) | | |
| 31 to 35 (n=72) | 34 (47.2) | 23 (31.9) | 15 (20.8) | | |
| 35 or older (n=52) | 18 (34.6) | 17 (32.7) | 17 (32.7) | | |
| **Number of children** | | | | | |
| One | 41 (53.9) | 23 (30.3) | 12 (15.8) | 8.26 | 0.083 |
| Two to three | 42 (40.8) | 34 (33) | 27 (26.2) | | |
| More than three | 9 (27.3) | 11 (33.3) | 13 (39.4) | | |
| **Mother's education** | | | | | |
| Uneducated | 0 | 1 (25) | 3 (75) | 33.85 | 0.001* |
| Literate | 11 (47.8) | 9 (39.1) | 3 (13.0) | | |
| Primary school | 4 (17.4) | 10 (43.9) | 9 (39.1) | | |
| Secondary school | 6 (42.9) | 4 (28.6) | 4 (28.6) | | |
| High school | 10 (23.8) | 23 (54.8) | 9 (21.4) | | |
| Bachelor or higher | 61 (57.5) | 23 (21.7) | 22 (20.8) | | |
| **Mother's job** | | | | | |
| Student | 1 (14.3) | 3 (42.9) | 3 (42.9) | 16.32 | 0.003* |
| Employee | 59 (56.2) | 24 (22.9) | 22 (21.0) | | |
| Housewife | 32 (32) | 43 (43) | 25 (25) | | |
| **Monthly income** | | | | | |
| Enough | 26 (65) | 8 (20) | 6 (15) | 10.25 | 0.036* |
| Partly enough | 31 (41.9) | 24 (32.4) | 19 (25.7) | | |
| Not enough | 35 (35.7) | 38 (38.8) | 25 (25.5) | | |
| **Smoker** | | | | | |
| Yes | 45 (45.9) | 30 (30.6) | 23 (23.5) | 0.58 | 0.740 |
| No | 47 (41.2) | 40 (35.1) | 27 (23.7) | | |
| **Region** | | | | | |
| Mount Lebanon | 27 (60) | 9 (20) | 9 (20) | 14.86 | 0.021* |
| Beqaa | 33 (32.7) | 41 (40.6) | 27 (26.7) | | |
| Beirut | 24 (51.1) | 11 (23.4) | 12 (25.5) | | |
| South | 8 (42.1) | 9 (47.4) | 2 (10.5) | | |

This table uses descriptive statistics: frequency (n) and percentage (%). Numbers with an asterisk correspond to significant associations.

with higher education degree (p=0.002) as well as those employed rather than housewives or students (p=0.003) (Table 3). In parallel, a significantly lower knowledge score was found in females with more children (p=0.006) and those that were non-smokers (p=0.04) (Table 3). No significant variations in knowledge scores were observed according to mother's age and income (Table 3).

Concerning the attitude, we found significantly higher attitude score with younger mother's (p=0.049) (Table 4). However, no significant variation in attitude was observed according to number of children, mother's education, job, income, smoking status and region (Table 4).

**Table 6. Multivariate analysis of sociodemographic and other factors associated with the KAP scores.**

| | Standardized Coefficient Beta | p= | 95% CI |
|---|---|---|---|
| **Model 1: Linear regression taking the knowledge score as the dependent variable** | | | |
| Mother's age | −0.12 | 0.088 | −0.56–0.04 |
| Mother's job | −0.21 | 0.002* | −1.42−−0.33 |
| Monthly income | −0.03 | 0.695 | −0.49-0.032 |
| **Model 2: Linear regression taking the attitude score as the dependent variable** | | | |
| Mother's age | −0.06 | 0.362 | −0.44–0.16 |
| Mother's job | 0.10 | 0.165 | −0.16–0.95 |
| Monthly income | −0.18 | 0.009* | −0.96−−0.13 |
| **Model 3: Linear regression taking the practice score as the dependent variable** | | | |
| Mother's age | −0.08 | 0.254 | −0.21–0.06 |
| Mother's job | −0.11 | 0.091 | −0.46–0.03 |
| Monthly income | −0.22 | 0.002* | −0.48−−0.11 |

The table presents adjusted odds ratios (OR) or hazard ratios (HR) with 95% confidence intervals (CI) and p-values with numbers with an asterisk corresponding to significant associations.

Lastly, practice scores varied significantly according to mother's job, income and region (Table 5), with better practice for employed mothers (p=0.003), as well as those that declared their income to be sufficient (p=0.036) and those residing in Mount Lebanon as compared to the other regions (p=0.021) (Table 5).

While the chi-square test determined if differences were statistically significant, the linear regression models tested the strength of relationships between variables. Multivariate analyses from three linear regression models were used to determine if mother's age, job or family income were significantly associated with KAP scores (Table 6). Results indicate that mother's employee status was significantly associated with higher knowledge score (95% CI=-1,42- -0.33, p=0.002), whereas mother's income was significantly associated with better attitude (95% CI=-0,96- -0.13, p=0.009) and practice (95% CI=-0,48- -0.11, p=0.002) scores (Table 6).

## Discussion

This cross-sectional study conducted among mothers of children aged between zero days to 5 years old aimed to assess the KAP regarding childhood immunization in Lebanon. It has revealed the need for targeted educational interventions and new governmental policies to improve vaccination practices and reduce the risk of outbreaks of preventable diseases. Overall, knowledge and attitude among surveyed mothers was excellent with respectively 86.3% and 94.8% of mothers presenting good scores. Interestingly, previous studies have shown disparities among regions in the level of health literacy, with a higher risk of having inadequate or problematic comprehensive health literacy (CHL) in people born in rural areas such as Beqaa [30]. This is in line with our results: a small number of mothers had poor knowledge about immunization, making up 8.5% of the sample and this issue was most prominent in Beqaa (11.9%). This suggests a need for more targeted outreach efforts in rural regions. Health disparities in Lebanon are influenced by a multitude of factors, including socioeconomic status, geographic location, and ongoing regional conflicts [31]. Healthcare access in the Beqaa and South Lebanon is notably limited compared to urban areas such as Beirut and Mount Lebanon. Previous studies have indicated that residents in rural regions face significant barriers to accessing quality healthcare services, which can be attributed to both economic constraints and the availability of healthcare facilities [12,32,33]. Furthermore, rurality often correlates with reduced access to quality education and fewer choices regarding educational pathways [34]. This disparity in access to education and healthcare underscores the challenges faced by populations in the Beqaa and South Lebanon, further entrenching socioeconomic inequalities. Recent studies indicate that the Beqaa region has faced outbreaks of infectious

diseases, such as hepatitis A and leishmaniasis, which have been exacerbated by the poor living conditions in informal tented settlements [24,25].

In a recent study, the knowledge and perception of Lebanese mothers regarding HPV vaccination for their children was determined. Socioeconomic factors, including education level and income, were also found to significantly influence mothers' willingness to vaccinate their children [23]. This is in line with our results showing that mother's income was significantly associated with better attitude (95% CI = -0,96- -0.13, p = 0.009) and practice (95% CI = -0,48- -0.11, p = 0.002) scores. Another interesting result is that younger mothers displayed significantly higher attitude score (p = 0.049). In Greece, maternal age was also positively correlated with favorable attitudes towards vaccination, suggesting that younger mothers, who are often more educated and informed, are likely to have better attitudes towards vaccination [35].

Compared to the overall good knowledge and attitude scores of participants, we found that only 43.3% of mothers presented a good practice score. Similarly, in a study involving mothers and fathers, only 32.8% of participants displayed a good practice score toward childhood immunization in Lebanon [22]. This is lower than our findings. The reason might be the exclusive participation of mothers in our study. In fact, it has been previously shown that mothers were more involved in the decision-making process of children's vaccination as mothers typically have more direct interactions with healthcare providers and are often the primary caregivers [36]. The discrepancy between knowledge and attitude v/s practice suggests that several barriers prevent mothers from translating their good knowledge and positive attitudes into action. These barriers could be related to accessibility, affordability, or sociocultural factors that vary across different regions. Interestingly, the analysis of participants' responses to the question "Do you feel that it is safe to have your child vaccinated?" revealed strong confidence in vaccine safety. A significant majority (94.3%) of respondents agreed that it is safe, while 4.2% were uncertain, and only 1.4% disagreed (S2 Fig). This indicates a high level of trust in vaccinations among the participants. We also found that a significant portion (36.3%) of respondents were uncertain about a link between vaccination and autism, while 54.2% disagreed with such a connection (S2 Fig). This indicates that while misinformation persists, the majority do not associate vaccines with autism. Parental hesitancy is also influenced by socio-demographic factors, including education level, income, and past experiences with vaccines. Research in Taiwan highlighted that parental hesitancy towards COVID-19 vaccination for children was associated with various factors, including parents' race, income, education level, and their perceptions of vaccine effectiveness and safety [37]. Similarly, a systematic review indicated that lower household income negatively impacted parents' willingness to vaccinate their children, suggesting that economic factors play a critical role in vaccination decisions [38]. Furthermore, it was recently shown that parents who had previously received vaccinations were more likely to vaccinate their children, indicating that personal experiences with vaccines could shape future decisions [39].To develop effective strategies and improve immunization coverage, it is necessary to understand the factors that affect KAP of mothers toward childhood immunization. We found that knowledge toward childhood immunization was associated with mother's occupation with employed mothers displaying better scores than housewives or students. Mother's income was significantly associated to better attitude and practice. Interestingly only 18.9% of mothers surveyed considered their income sufficient. This is the direct reflect of the economic crisis that started in 2019 and has affected the Lebanese healthcare system [21].

Concerning factors affecting practice scores, we found that mother's job, income and region were significantly associated to practice scores. In particular, significant variations were observed for practice score according to region, with more than 60% of mothers living in Mount Lebanon and Beirut exhibiting good practice scores, while this level fell to 32.7% in Beqaa. This is in line with the study by Mansour et al., showing that some regions in Lebanon presented vaccination coverage rates up to almost 100%, while others presented rates as low as 20.1% [40]. Other factors that might negatively affect mothers' practice toward childhood immunization include the unavailability and high cost of vaccines. This is particularly true in challenging socio-economic situations. In Somalia, it was found that hepatitis B vaccination coverage was low due to the unavailability and high cost of vaccines [41]. Similarly, in Japan, household income was associated with high vaccine coverage, indicating that cost is a barrier for varicella immunization [42]. Recognition of the socioeconomic

barriers to immunization is therefore necessary to promote better childhood immunization practice. Another way to improve vaccination practice is to promote optional vaccines such as rotavirus and influenza vaccines. In our study, only 49.1% of mothers reported that their children had received optional vaccines (S2 Fig). Although the Advisory Committee on Immunization Practices (ACIP) recommends yearly influenza vaccination for children starting 6 months old, worldwide data estimate that percentage of children vaccinated against influenza varies according to age and country [43]. Lastly, lack of recommendation of optional vaccines by physicians is a common cited barrier to better vaccination practice. In a recent study by Zakhour et al., it was the most stated cause for the absence of children influenza vaccination among parents in greater Beirut region [44]. The importance of healthcare workers and physicians in building trust about vaccines with parents is essential to improve the use of optional vaccines [22,45].

Among mothers who had knowledge on how to boost children's immune response, 84.4% believed that healthy nutrition was the most important way. This is particularly relevant as lower antibody responses to Hepatitis B [46], measles [47], pertussis [48], and tetanus [49] vaccination have been reported in malnourished children. Other common answers included breastfeeding (17.7%) with 76.25% of surveyed mothers reporting to have breastfed their child. The beneficial effect of breastfeeding on children immunization has been proven more than 30 years ago by comparing antibody levels following vaccination with poliovirus, diphtheria and tetanus toxoids and *Haemophilus influenzae* type b vaccine in breast-fed and formula-fed children [50,51]. In Lebanon, breastfeeding prevalence was estimated at 67% at 8–12 weeks postpartum [52], but because the study focused exclusively on an urban population of first-time mothers in Beirut, results may not reflect the regional disparities affecting breastfeeding prevalence. In fact, in our study a higher prevalence of breastfeeding was found (76.25% of respondents) with 6–12 months being the most reported breastfeeding duration (36.7%) and 76.7% of mothers breastfeeding for at least 6 months. This implies that Lebanese mothers are aware of the importance of breastfeeding and that most of them follow the WHO recommendation to exclusive breastfeed for at least the first 6 months of life. Regional disparities related to breastfeeding practices in urban versus rural regions exist. It was found that initiation rates of breastfeeding could reach 90% in rural communities compared to 76% in urban areas, highlighting a notable disparity in breastfeeding behaviors [53].

In conclusion, this study gathering data from mothers residing in Mount Lebanon, Beirut, South and Beqaa, encompassed a substantial portion of the Lebanese population. This broad representation enhances the generalizability of the findings. However, there are several limitations to our study. First, the use of a cross-sectional design restricts the ability to establish causal relationships between mothers' KAP and children's immunization status. Second, because traditional income measures often fail to capture the nuances of household economic instability [54], we used self-reported income data. Despite its benefit in the context of our study, it may be subject to social desirability bias. Future research may consider alternative economic indicators, such as food security levels or specific cost-of-living indices, to complement income self-reporting. Another limitation is the use of a 3-point Likert scale for calculation of KAP scores. While this may limit the ability to capture nuanced variations in opinions compared to a broader Likert scale, it minimizes indecisiveness and ensures consistent interpretation across respondents. Lastly, the study did not account for the influence of healthcare providers on maternal decisions regarding vaccination. Despite these limitations, our findings provide valuable insights into the challenges of childhood immunization in Lebanon and highlight key areas for targeted interventions. Based on the findings of this study, increased efforts to reach mothers with low education level are needed, especially in rural regions. Education should be tailored to the mothers' level of education and should be delivered through trusted sources of information. Additionally, regular monitoring and evaluation of immunization programs to identify any gaps are necessary. For instance, Yazdani et al. emphasize the importance of addressing implementation barriers to routine immunization, particularly in peri-urban settings where mothers may have lower educational attainment [54]. This aligns with the findings of Karami et al., who stress that monitoring vaccination coverage is a vital component of immunization programs, indicating that educational efforts should be coupled with robust data collection and analysis to identify areas needing improvement [55]. In another study, the use of SMS reminders for routine immunization in Northern Nigeria, which not only aids in

reminding parents but also serves as a performance monitoring tool for health workers was evaluated. Results showed that integrating technology with educational efforts could enhance the effectiveness of immunization programs, particularly in rural areas where access to information may be limited [56].

Lastly, developing new policies to address the need for global harmonization of routine vaccination in Lebanon and to increase governmental financial support will help improve childhood vaccination coverage in the country. The role of international partners such as WHO and UNICEF, as well as local community leaders and organizations, in facilitating the successful rollout of childhood vaccines has been documented in several countries [57]. This underscores the importance of collaborative efforts and tailored policies to address specific local needs.

Further research is needed to better assess the post-crisis challenges and remaining barriers toward childhood immunization in Lebanon. Improving KAP towards childhood immunization is necessary for Lebanon to reach the target immunization coverage recommended by the WHO [58].

## Supporting information

**S1 Fig. The questionnaire (English version).**
(PDF)

**S2 Fig. Additional questions on vaccination.**
(TIF)

**S3 Fig. The raw data file.**
(XLSX)

## Acknowledgments

The authors would like to thank all the participants.

## Author contributions

**Conceptualization:** Pia Chedid.

**Formal analysis:** Pia Chedid.

**Investigation:** Sara Saleh.

**Methodology:** Sara Saleh.

**Supervision:** Pia Chedid.

**Writing – original draft:** Sara Saleh.

**Writing – review & editing:** Pia Chedid.

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
