## [Decision Letter · Decision Letter 0]

30 Dec 2024

PONE-D-24-37390Knowledge, attitude and practice toward childhood immunization among mothers in LebanonPLOS ONE

Dear Dr. Chedid,

Thank you for submitting your manuscript to PLOS ONE. After careful consideration, we feel that it has merit but does not fully meet PLOS ONE’s publication criteria as it currently stands. Therefore, we invite you to submit a revised version of the manuscript that addresses the points raised during the review process.

We look forward to receiving your revised manuscript.

Kind regards,

dR. Walid Al-Shaar

Academic Editor

PLOS ONE

Journal Requirements:

2. In the online submission form, you indicated that [Data are available on request from the author.].

Additional Editor Comments (if provided):

Reviewers' comments:

Reviewer's Responses to Questions

**Comments to the Author**

1. Is the manuscript technically sound, and do the data support the conclusions?

Reviewer #1: No

Reviewer #2: Yes

2. Has the statistical analysis been performed appropriately and rigorously? 

Reviewer #1: No

Reviewer #2: No

3. Have the authors made all data underlying the findings in their manuscript fully available?

Reviewer #1: No

Reviewer #2: Yes

4. Is the manuscript presented in an intelligible fashion and written in standard English?

Reviewer #1: No

Reviewer #2: No

5. Review Comments to the Author

Reviewer #1: Reviewer Report for Manuscript PONE-D-24-37390

Title: Knowledge, Attitude, and Practice Toward Childhood Immunization Among Mothers in Lebanon

This manuscript addresses a critical area of public health—childhood immunization coverage among Lebanese mothers—through a cross-sectional survey assessing their knowledge, attitudes, and practices (KAP). However, while the topic is relevant, the manuscript in its current form lacks sufficient rigor in terms of methodological clarity, depth of analysis, and actionable conclusions. Significant revisions are required to address gaps in methodology, deepen analysis, and provide actionable recommendations. Below are major concerns across each section that, if addressed, could strengthen the manuscript.

First, the topic of KAP toward childhood immunization among mothers in Lebanon is not entirely new, as there have been previous studies examining vaccination coverage, parental attitudes, and factors influencing immunization practices in Lebanon. Elaborate on the rationale of conducting such as study.

Abstract

This abstract has significant weaknesses that reduce from its clarity and effectiveness in conveying the study's scientific value.

1. Generic and Weak Background:

o The opening is vague and provides no insight into the specific context of childhood immunization in Lebanon. Simply stating that immunization coverage is "suboptimal in some countries, including Lebanon" does not capture the urgency or relevance of this topic.

o The phrase "knowledge, attitudes, and practices (KAP) of mothers play a key role" is too generic and does not explain why this study is specifically focusing on mothers or what gaps it addresses in previous research.

2. Poorly Defined Study Aim:

3. The methods section is vague, providing only the type of study (cross-sectional) and basic details about the participants (mothers with children aged 0-5). There is no mention of the sampling method, location details which raises questions about the validity and reliability of the data. Statistical tests are mentioned (linear regression), but their relevance to the study objectives is unclear.

4. Incoherent Presentation of Results:

o The results section is cluttered with fragmented data points that feel disjointed.

o The associations between KAP and specific sociodemographic factors are presented with wrong way.

5. Weak Conclusion with Little Practical Insight

6. This abstract is more like a collection of scattered facts than a well-integrated abstract of a manuscript.

Introduction

This introduction is weak and lack clearer focus and stronger rationale for the study.

1. The introduction starts by discussing the Under-Five Mortality Rate (U5MR), but it doesn’t connect this topic clearly to the study's focus on maternal KAP regarding childhood immunization in Lebanon. While U5MR is relevant, it could be introduced more concisely and connected more explicitly to the importance of vaccination and maternal influence on immunization coverage.

2. Additionally, background information about global vaccination benefits (such as DPT3’s effect on U5MR) is overly generalized. This section should be streamlined to focus specifically on vaccination in Lebanon, especially given the current public health challenges in the country.

3. Although the socioeconomic crisis in Lebanon is briefly mentioned, it deserves more emphasis, as it profoundly impacts healthcare access and vaccination rates.

4. What about the routine vaccination for children in Lebanon. You must mention the routine immunization calendar adopted by MOPH, the provision of free vaccines through PHC. Expanding on the unique barriers faced by Lebanese mothers—such as economic hardship, vaccine availability, and healthcare system strain—would make the rationale for this study more compelling and specific to the local context??

5. The mention of a past measles outbreak and increased mumps cases is relevant, but it should be integrated more effectively to underscore Lebanon's vulnerability in terms of vaccination coverage. More information is needed.

6. Check the Epidemiological surveillance unit reports on Vaccine preventable diseases including information about the age group affected by these outbreaks.

7. Check the vaccination coverage from PHC and EPI at MOPH.

8. The introduction should provide a more thorough review of existing studies on immunization attitudes and practices in Lebanon to clarify the gaps this study aims to address.

9. The authors mention that “few studies have assessed the KAP of mothers toward childhood immunization in Lebanon,” but they do not specify what those studies have found or how this study builds on or differs from them.

10. The authors do not clearly justify why this KAP study is needed now, especially given recent socioeconomic challenges. Highlighting how this study aims to capture current, potentially evolving attitudes and practices under these economic pressures would strengthen the rationale.

11. The introduction should also clarify why focusing on mothers, specifically, is critical. While the role of mothers in ensuring childhood immunization is noted, it could be expanded to explain how their attitudes and practices uniquely impact vaccination rates.

12. A clear research question or hypothesis is missing, leaving readers without a concrete understanding of the study’s aims.

13. Some parts of the introduction contain redundant information, such as repeated mentions of vaccination benefits and general global vaccination data, which could be streamlined to avoid redundancy.

Methods

This Materials and Methods section, has several issues that detract from its rigor, and methodological transparency.

1. Lack of Detail in Sampling Strategy: The statement that participants were "enrolled randomly" is vague and lacks critical detail. The authors should explain the exact randomization process used, particularly how they ensured equal representation across the different Lebanese regions.

2. Sampling Bias Risk: There is no mention of how the study addressed or minimized sampling bias, particularly since different regions in Lebanon might have varying accessibility. Without further clarification, it’s unclear how representative this sample truly is of Lebanese mothers.

3. Study Duration: Although the dates are specified (July 5 to September 27, 2023), there’s no justification for the study period. The authors should clarify if this period aligns with any particular relevance, such as vaccine campaign schedules, which could influence maternal attitudes or behaviors.

4. The description of how participants were invited is vague. The authors state that in-person interviews were conducted by trained personnel, but there’s no information on how or where these participants were approached (e.g., hospitals, clinics, community centers).

5. Inclusion/Exclusion Criteria Overlook Details

6. There is no mention of whether the questionnaire was validated or pilot-tested prior to the study. This raises questions about its reliability and whether it accurately captures the constructs of knowledge, attitude, and practice. The authors should state if the questionnaire was adapted from a validated tool or describe any validation procedures if it was newly developed.

7. The questionnaire's structure description is superficial, especially in sections like “Knowledge Assessment” and “Practice Assessment.” Each section could benefit from clearer definitions of terms like “confidence in vaccination” or “up-to-date vaccination practices,” which may vary across participants.

8. The use of a 3-point Likert scale ("agree," "uncertain," and "disagree") is simplistic and may not capture the complexity of participants' attitudes and practices. A broader scale would provide more nuanced data, particularly for a study measuring sensitive issues like health beliefs and practices.

9. The authors mention that total KAP scores were categorized into "poor," "moderate," and "good" but they don’t explain how these categories were defined or what thresholds were used. This lack of transparency raises concerns about the validity of the KAP groupings and whether they accurately reflect participant responses.

5. Statistical Analysis

• The statistical methods are insufficiently detailed. there is no clear explanation of why these methods were chosen or how they were applied.

• There is no information on whether the authors controlled for potential confounding variables in their analyses. Without this, the results could be misleading.

• Questionnaire Design:

o The questionnaire structure is briefly described, but there is no information on its validation or any previous testing. If the questionnaire was adapted from existing tools, references should be provided, and any modifications noted.

Results:

The results section is also weak.

• Inadequate Analysis of KAP Gaps:

• Over-Simplified Categorization:

• Inconsistent Regional Comparisons:

…..

Discussion

• Lack of In-Depth Interpretation:

• Weak Link to Practical Recommendations

• Insufficient Acknowledgment of Study Limitations

Conclusion

The conclusion merely summarizes the findings without offering a strong takeaway or specific call to action.

Minor Comments

1. Clarity in Terminology: Terms like “good” and “poor” could be made more precise by referring to the actual scoring criteria.

2. Ensure that all tables and figures are self-explanatory and fully annotated to aid readers in understanding the data without referring back to the text.

3. A clear statement on the data availability would enhance transparency and replicability.

Recommendation: Reject

Reviewer #2: Key Points and Suggestions for Improvement:

1. Regional Discrepancies in Sociodemographic Factors: While variations between regions in mothers' education, job, and income were noted, the manuscript could better explain potential reasons behind these regional disparities, providing more context or references.

2. Sample Size and Methodology: The absence of a detailed sample size estimation and explanation of sampling methodology is a limitation. Including this information would strengthen the study's validity and address the query on stratified sampling across regions.

3.Categorization of Scores: Clarification is needed regarding the categorization criteria for KAP (Knowledge, Attitude, Practice) scores into poor, moderate, and good. Details on thresholds and visual binning in SPSS should be included for transparency.

4. Income Classification: The manuscript mentions income categories (enough, partly enough, not enough) without defining the criteria. Providing specific thresholds or explaining these classifications will enhance clarity.

5. Definition of Literacy: The definition of "literate" as used in the study is unclear. Explicitly stating the criteria for literacy classification would be beneficial.

6. Barriers to Immunization Practices: The discrepancy between good knowledge/attitude and lower practice scores suggests barriers. These could be better elaborated upon, potentially including cultural, logistical, or economic factors.

7. Breastfeeding and Nutrition Insights:The study highlights breastfeeding practices and their impact on immunity. While this is informative, adding comparative insights or referencing breastfeeding rates from similar contexts would enrich the discussion.

8. Policy Recommendations: The manuscript’s conclusion about the need for educational interventions and policy support could be expanded. Specific strategies or examples from similar socioeconomic contexts could offer actionable insights.

9. Use of Multivariate Analysis:While the multivariate analysis identifies significant associations, presenting odds ratios and confidence intervals for key variables would clarify the relationships.

6. PLOS authors have the option to publish the peer review history of their article (what does this mean? ). If published, this will include your full peer review and any attached files.

**Do you want your identity to be public for this peer review?** For information about this choice, including consent withdrawal, please see our Privacy Policy .

Reviewer #1: **Yes: ** Dr. Dalal Youssef

Reviewer #2: **Yes: ** Dr. Ayushi Agrawal

---

## [Author Response · Author response to Decision Letter 1]

26 Feb 2025

Answers to Reviewer #1

Reviewer #1: Reviewer Report for Manuscript PONE-D-24-37390

Title: Knowledge, Attitude, and Practice Toward Childhood Immunization Among Mothers in Lebanon

This manuscript addresses a critical area of public health—childhood immunization coverage among Lebanese mothers—through a cross-sectional survey assessing their knowledge, attitudes, and practices (KAP). However, while the topic is relevant, the manuscript in its current form lacks sufficient rigor in terms of methodological clarity, depth of analysis, and actionable conclusions. Significant revisions are required to address gaps in methodology, deepen analysis, and provide actionable recommendations. Below are major concerns across each section that, if addressed, could strengthen the manuscript.

First, the topic of KAP toward childhood immunization among mothers in Lebanon is not entirely new, as there have been previous studies examining vaccination coverage, parental attitudes, and factors influencing immunization practices in Lebanon. Elaborate on the rationale of conducting such as study.

Answer: As suggested, we elaborated on the rationale of the study by focusing on the unique barriers faced by Lebanese mothers and the importance of capturing evolving attitudes and practices under economic pressures, to amend the introduction of the manuscript lines 83-85, lines 107-110 and lines 128-134.

Abstract

This abstract has significant weaknesses that reduce from its clarity and effectiveness in conveying the study's scientific value.

Answer: We have modified the abstract according to the reviewer’s comments. However, because the number of words in the abstract is limited to 300 words, some concerns have been addressed in the Methods or Discussion section of the article.

1. Generic and Weak Background:

o The opening is vague and provides no insight into the specific context of childhood immunization in Lebanon. Simply stating that immunization coverage is "suboptimal in some countries, including Lebanon" does not capture the urgency or relevance of this topic.

Answer: We have now elaborated on urgency and relevance of the topic lines 22-24.

o The phrase "knowledge, attitudes, and practices (KAP) of mothers play a key role" is too generic and does not explain why this study is specifically focusing on mothers or what gaps it addresses in previous research.

Answer: We have elaborated in the introduction on the key role of mothers in childhood immunization lines 116-119.

2. Poorly Defined Study Aim:

Answer: We have clarified the study aim in the abstract line 28.

3. The methods section is vague, providing only the type of study (cross-sectional) and basic details about the participants (mothers with children aged 0-5). There is no mention of the sampling method, location details which raises questions about the validity and reliability of the data. Statistical tests are mentioned (linear regression), but their relevance to the study objectives is unclear.

Answer: We have amended the abstract with details on the sampling method line 30. The other comments have been answered in the methods section.

4. Incoherent Presentation of Results:

o The results section is cluttered with fragmented data points that feel disjointed.

o The associations between KAP and specific sociodemographic factors are presented with wrong way.

Answer: The results were reformulated in a concise manner to highlight the specific sociodemographic factors and their association with knowledge, attitude and practice scores lines 34-41.

5. Weak Conclusion with Little Practical Insight

The conclusion has now been revised lines 49-52.

6. This abstract is more like a collection of scattered facts than a well-integrated abstract of a manuscript.

Answer: We believe the revised abstract now captures the relevance and novelty of the results.

Introduction

This introduction is weak and lack clearer focus and stronger rationale for the study.

Answer: We believe the revised Introduction now clearly presents the background and rationale for the study.

1. The introduction starts by discussing the Under-Five Mortality Rate (U5MR), but it doesn’t connect this topic clearly to the study's focus on maternal KAP regarding childhood immunization in Lebanon. While U5MR is relevant, it could be introduced more concisely and connected more explicitly to the importance of vaccination and maternal influence on immunization coverage.

2. Additionally, background information about global vaccination benefits (such as DPT3’s effect on U5MR) is overly generalized. This section should be streamlined to focus specifically on vaccination in Lebanon, especially given the current public health challenges in the country.

Answer to 1. and 2.: We have now amended the Introduction lines 75-77 and 83-85 to present in a concise manner the link between U5MR and maternal KAP with focus on the Lebanese context.

3. Although the socioeconomic crisis in Lebanon is briefly mentioned, it deserves more emphasis, as it profoundly impacts healthcare access and vaccination rates.

Answer: We have now amended the Introduction lines 128-134 to present the profound impact of the socioeconomic crisis on healthcare access and vaccination rates in Lebanon.

4. What about the routine vaccination for children in Lebanon. You must mention the routine immunization calendar adopted by MOPH, the provision of free vaccines through PHC.

Answer: We have now amended the Introduction lines 135-140 to present the Lebanese routine immunization calendar.

Expanding on the unique barriers faced by Lebanese mothers—such as economic hardship, vaccine availability, and healthcare system strain—would make the rationale for this study more compelling and specific to the local context??

Answer: We have now amended the Introduction lines 102-110 to present the unique barriers faced by Lebanese mothers, especially in rural Lebanese regions.

5. The mention of a past measles outbreak and increased mumps cases is relevant, but it should be integrated more effectively to underscore Lebanon's vulnerability in terms of vaccination coverage. More information is needed.

6. Check the Epidemiological surveillance unit reports on Vaccine preventable diseases including information about the age group affected by these outbreaks.

7. Check the vaccination coverage from PHC and EPI at MOPH.

Answer to 5. And 6.: Although recent measles and mumps outbreaks in Lebanon are indicators of a insufficient children immunization coverage, we believe it would be out of the scope of our study to discuss it in more details. It would be suitable for an epidemiology surveillance study , which is not the case.

8. The introduction should provide a more thorough review of existing studies on immunization attitudes and practices in Lebanon to clarify the gaps this study aims to address.

Answer: We thank the reviewer for this important point. We have now provided more insights into the existing studies in Lebanon lines 121-127.

9. The authors mention that “few studies have assessed the KAP of mothers toward childhood immunization in Lebanon,” but they do not specify what those studies have found or how this study builds on or differs from them.

Answer: We thank the reviewer for this important comment. Literature on maternal knowledge, attitude and practice toward childhood immunization is in general scarce, and particularly in Lebanon. We have however updated our Introduction lines 121-127 and Discussion by highlighting recent articles such as a study published in 2024 on knowledge and perception of Lebanese mothers toward HPV vaccination (lines 322-329).

Elissa N, Charbel H, Marly A, Ingrid N, Nadine S, Rachel A. Knowledge and perception of HPV vaccination beamong Lebanese mothers of children between nine and 17 years old. Reprod Health. 2024 Mar 27;21(1):40. doi: 10.1186/s12978-024-01764-7. PMID: 38539219; PMCID: PMC10967097.

10. The authors do not clearly justify why this KAP study is needed now, especially given recent socioeconomic challenges. Highlighting how this study aims to capture current, potentially evolving attitudes and practices under these economic pressures would strengthen the rationale.

Answer: We have highlighted this important aspect of the study to strengthen the rationale lines 141-147.

11. The introduction should also clarify why focusing on mothers, specifically, is critical. While the role of mothers in ensuring childhood immunization is noted, it could be expanded to explain how their attitudes and practices uniquely impact vaccination rates.

Answer: We have now developed the particular role of mothers in ensuring children receive proper vaccination lines 116-119.

12. A clear research question or hypothesis is missing, leaving readers without a concrete understanding of the study’s aims.

Answer: Rather than testing a specific hypothesis, the study determined maternal KAP toward childhood immunization and investigated multiple factors, allowing for a more comprehensive understanding of the barriers affecting vaccination practices. This is now clearly stated lines 147-149.

13. Some parts of the introduction contain redundant information, such as repeated mentions of vaccination benefits and general global vaccination data, which could be streamlined to avoid redundancy.

Answer: We believe that with the adjustments made, the Introduction now present the background and aim of the study without redundancy.

Methods

This Materials and Methods section, has several issues that detract from its rigor, and methodological transparency.

1. Lack of Detail in Sampling Strategy: The statement that participants were "enrolled randomly" is vague and lacks critical detail. The authors should explain the exact randomization process used, particularly how they ensured equal representation across the different Lebanese regions.

Answer: Sample size estimation and methodology details have been included in the revised methods section lines 161-169.

2. Sampling Bias Risk: There is no mention of how the study addressed or minimized sampling bias, particularly since different regions in Lebanon might have varying accessibility. Without further clarification, it’s unclear how representative this sample truly is of Lebanese mothers.

Answer: We thank the reviewer for this important comment. We have now elaborated on this point lines 170-171.

3. Study Duration: Although the dates are specified (July 5 to September 27, 2023), there’s no justification for the study period. The authors should clarify if this period aligns with any particular relevance, such as vaccine campaign schedules, which could influence maternal attitudes or behaviors.

Answer: The study follows a cross-sectional design, which captures data at a single point in time rather than assessing changes over a period. Moreover, the choice of study duration does not require a specific justification, as it does not influence the validity of the findings. In fact, maternal KAP toward childhood immunization are unlikely to be affected by seasonal variations or vaccine campaign schedules. These factors generally shape long-term perceptions rather than causing short-term fluctuations. Consequently, the selected timeframe remains appropriate for assessing the target variables without introducing seasonal bias.

4. The description of how participants were invited is vague. The authors state that in-person interviews were conducted by trained personnel, but there’s no information on how or where these participants were approached (e.g., hospitals, clinics, community centers).

Answer: The methods section has been amended to answer this question lines 170-171.

5. Inclusion/Exclusion Criteria Overlook Details

Answer: The inclusion and exclusion criteria were carefully defined in the Methods section to ensure the study’s relevance and validity. We included Lebanese mothers with at least one child aged between zero and five years to align with the study's focus on early childhood immunization. The ability to comprehend and respond in Arabic or English was specified to ensure accurate data collection through in-person interviews.

The exclusion criteria were also clearly outlined to eliminate potential confounders. Non-Lebanese mothers or those not residing in Lebanon were excluded to maintain a homogeneous sample representative of the Lebanese population. Similarly, mothers without children in the specified age range were excluded as their experiences would not be relevant. Finally, unwilling participants or those unable to communicate effectively in Arabic or English were excluded to ensure data reliability.

6. There is no mention of whether the questionnaire was validated or pilot-tested prior to the study. This raises questions about its reliability and whether it accurately captures the constructs of knowledge, attitude, and practice. The authors should state if the questionnaire was adapted from a validated tool or describe any validation procedures if it was newly developed.

Answer: Thank you for your comment. The questionnaire utilized in this study was developed by incorporating items from a previously validated survey by Almutairi et al. (2021), which specifically assessed mothers’ KAP regarding childhood vaccination during the first five years of life. Given that the questionnaire was adapted from an established tool, its content validity is supported. Additionally, the English version of the full questionnaire has been uploaded as S1 Fig for further review, and the methods section has been amended lines 183 and 185.

7. The questionnaire's structure description is superficial, especially in sections like “Knowledge Assessment” and “Practice Assessment.” Each section could benefit from clearer definitions of terms like “confidence in vaccination” or “up-to-date vaccination practices,” which may vary across participants.

Answer: The full questionnaire has now been uploaded as S1 Fig for further review and the Methods section has been amended to include number of items per section relevant (lines 187, 194, 197 and 200).

8. The use of a 3-point Likert scale ("agree," "uncertain," and "disagree") is simplistic and may not capture the complexity of participants' attitudes and practices. A broader scale would provide more nuanced data, particularly for a study measuring sensitive issues like health beliefs and practices.

Answer: Thank you for this comment. We have now mentioned the limits of the 3-point Likert scale in the discussion lines 417-419. However, we’d like to point out that using a 3-point Likert scale was deliberately chosen to simplify responses, reduce respondent burden, and enhance clarity, particularly in our diverse population that included participants from lower educational levels. A study comparing the time required to complete surveys using different response options, including a 3-point Likert scale have shown that a 3-point scale is quicker to complete and provides results that are comparable to those obtained from a 5-point scale (Jeong et al., 2016).

Jeong, H., Park, M., Kim, C., & Lee, W. (2016). Untitled. Biometrics & Biostatistics International Journal, 4(5). https://doi.org/10.15406/bbij.4.5

9. The authors mention that total KAP scores were categorized into "poor," "moderate," and "good" but they don’t explain how these categories were defined or what thresholds were used. This lack of transparency raises concerns about the validity of the KAP groupings and whether they accurately reflect participant responses.

Answer: Th methods section has been amended to answer this question lines 208-209.

5. Statistical Analysis

• The statistical methods are insufficiently detailed. there is no clear explanation of why these methods were chosen or how they were applied.

• There is no information on whether the authors controlled for potential confounding variables in their analyses. Without this, the results could be misleading.

Answer: Thank you for your comment. The statistical analysis for this study was conducted using SPSS version 24 for Windows. A significance level of p < 0.05 was considered statistically significant. To assess associations between categorical variables, we used the chi-square test, which is appropriat

---

## [Editor Report · Decision Letter 1]

18 Mar 2025

Knowledge, attitude and practice toward childhood immunization among mothers in Lebanon

PONE-D-24-37390R1

Dear Dr. Chedid,

We’re pleased to inform you that your manuscript has been judged scientifically suitable for publication and will be formally accepted for publication once it meets all outstanding technical requirements.

Kind regards,

Dr. Walid Al-Shaar

Academic Editor

PLOS ONE

---

## [Editor Report · Acceptance letter]

PONE-D-24-37390R1

PLOS ONE

Dear Dr. Chedid,

I'm pleased to inform you that your manuscript has been deemed suitable for publication in PLOS ONE. Congratulations! Your manuscript is now being handed over to our production team.

Kind regards,

on behalf of

Dr. Walid Al-Shaar

Academic Editor

PLOS ONE